# Shallow-level defect passivation by 6H perovskite polytype for highly efficient and stable perovskite solar cells

Hobeom Kim [1,2,10] ✉, So-Min Yoo[1,3,10], Bin Ding [1,10], Hiroyuki Kanda [1], Naoyuki Shibayama [4], Maria A. Syzgantseva[5,6], Farzaneh Fadaei Tirani [1], Pascal Schouwink[1], Hyung Joong Yun[7], Byoungchul Son[8], Yong Ding [1], Beom-Soo Kim[3], Young Yun Kim [3], Junmo Park[2], Olga A. Syzgantseva [5], Nam Joong Jeon [3] ✉, Paul J. Dyson [1] ✉ & Mohammad K. Nazeeruddin [1,9] ✉

The power conversion efficiency of perovskite solar cells continues to increase. However, defects in perovskite materials are detrimental to their carrier dynamics and structural stability, ultimately limiting the photovoltaic characteristics and stability of perovskite solar cells. Herein, we report that 6H polytype perovskite effectively engineers defects at the interface with cubic polytype $FAPbI_3$, which facilitates radiative recombination and improves the stability of the polycrystalline film. We particularly show the detrimental effects of shallow-level defect that originates from the formation of the most dominant iodide vacancy ($V_I^+$) in $FAPbI_3$. Furthermore, additional surface passivation on top of the hetero-polytypic perovskite film results in an ultra-long carrier lifetime exceeding 18 µs, affords power conversion efficiencies of 24.13% for perovskite solar cells, 21.92% (certified power conversion efficiency: 21.44%) for a module, and long-term stability. The hetero-polytypic perovskite configuration may be considered as close to the ideal polycrystalline structure in terms of charge carrier dynamics and stability.

Perovskite solar cells (PSCs) have excellent power conversion efficiencies (PCEs) due to the excellent optoelectronic properties of the carefully engineered metal-halide perovskite light absorbers[1–4]. The state-of-the-art PSCs with the highest PCEs are mostly fabricated from three-dimensional (3D) polycrystalline formamidinium lead iodide ($FAPbI_3$) due to its narrow bandgap ($E_g$) and thermal stability with a relatively high formation energy[5–9]. Nevertheless, polycrystalline perovskites suffer from defects that cause loss of charge carriers and

degradation of structural stability, thus limiting photovoltaic efficiency and device stability. More specifically, deep-level defects are certainly regarded as undesirable because they are the primary non-radiative recombination centers. In contrast, the benign nature of shallow-level defects i.e., the so-called defect tolerance of perovskites, has been generally known to afford the high-efficiency PSCs based on the idea that they enable efficient radiative recombination of charge carriers[10]. However, shallow-level defects such as the dominant iodide

[1]Institute of Chemical Sciences and Engineering, École Polytechnique Fédérale de Lausanne (EPFL), Lausanne CH-1015, Switzerland. [2]School of Materials Science and Engineering, Gwangju Institute of Science and Technology (GIST), Gwangju 61005, Republic of Korea. [3]Division of Advanced Materials, Korea Research Institute of Chemical Technology (KRICT), Daejeon 34114, Republic of Korea. [4]Faculty of Biomedical Engineering, Graduate School of Engineering, Toin University of Yokohama, Yokohama 225-8503, Japan. [5]Department of Chemistry, Lomonosov Moscow State University, Moscow 119991, Russia. [6]Department of Physics, Mendeleev University of Chemical Technology, Moscow 125047, Russia. [7]Research Center for Materials Analysis, Korea Basic Science Institute (KBSI), Daejeon 34133, Republic of Korea. [8]Center for Research Equipment, Korea Basic Science Institute (KBSI), Daejeon 34133, Republic of Korea. [9]Center of Excellence for Advanced Materials Research (CEAMR), King Abdulaziz University, Jeddah 21589, Saudi Arabia. [10]These authors contributed equally: Hobeom Kim, So-Min Yoo, Bin Ding. ✉e-mail: hobkim@gist.ac.kr; njjeon@krict.re.kr; paul.dyson@epfl.ch; mdkhaja.nazeeruddin@epfl.ch

vacancy ($V_I^+$) may be significantly problematic as they can lead to the formation of polarons that activate non-radiative processes, eventually degrading the photovoltaic performance of PSCs[11]. Although recent studies have started to address the concern of shallow-level defects more than before, their focus is mainly concerned with their migration due to the ionic nature, rather than their impact on carrier dynamics[12–14]. Thus, a comprehensive understanding of the impact of the shallow-level defects on charge carrier recombination dynamics in perovskites remains elusive, and we suggest a reappraisal of the idea of the benignancy of shallow-level defects in perovskites.

In general, defect engineering is required to improve the quality of the surface or bulk of polycrystalline perovskite films. The introduction of a surface passivation layer on top of the perovskite film has become a standard method to reduce surface defects and suppress undesirable non-radiative recombination, increase moisture resistance, and thus enhance device performance[15–18]. In particular, low-dimensional perovskites (LDPs), e.g. 2D perovskite or quasi-2D perovskite, are often highly compatible with the underlying 3D perovskite[17], and the thermodynamically stable LDPs with higher formation energies based on strong Van der Waals interactions between the cations can prevent decomposition of the underlying 3D perovskite to improve both film and device stability[19,20]. In contrast, bulk defect passivation can be rather challenging. A typical approach is to introduce an external chemical reagent, such as a bulky ammonium cation or an organic passivator to passivate the defects[21,22]. However, the external reagent (additive or dopant) can directly impact the crystalline quality of the perovskite during the crystal growth as the structural integrity of the perovskite may be degraded. The use of LDPs represents an advanced strategy to maintain the material homogeneity as they belong to the same materials category of metal-halide perovskite[21]. Nevertheless, the constituents of LDPs differ from those of the 3D perovskites and the bulky, insulating organic cations in LDPs limits charge transport within the 3D perovskite film.

Unlike previous strategies, we demonstrate an approach that creates a new sector of defect passivation which involves the exploitation of perovskite polytypes. As polytypes consist of identical constituents but only differ in the atomic stacking sequence, employment of desirable polytypes may provide a way to achieve the highest crystallinity of polycrystalline perovskites while maintaining homogeneity and integrity of the material based on high lattice coherency between the polytypes. Here, we report that the incorporation of a hexagonal polytype (6H) perovskite into the bulk of FAPbI$_3$ effectively engineers the defects of $\alpha$-phase cubic polytype (3 C) of FAPbI$_3$. Unlike the 100% face-sharing hexagonal polytype (2H) of FAPbI$_3$[23], the 6H polytype, which includes a 66% corner-sharing component[24], can coherently intervene in the dominant $V_I^+$ defect site at the surface of the 3C-FAPbI$_3$ grain, thus screening the defects and bridging the cubic phases. We also unveil a complex nonradiative recombination process initiated by the formation of the dominant shallow-level defect $V_I^+$. The inclusion of the 6H phase improves structural integrity and imparts better stability to the polycrystalline perovskite films. Thus, the hetero-polytypic perovskite consisting of the 3 C and a hexagonal polytype with its corner-sharing component may be considered as having close to the ideal structure for a polycrystalline perovskite film to maintain material homogeneity, suppress non-radiative processes and enhance stability. Furthermore, the introduction of an additional LDP passivation layer on top of the 3 C/6H hetero-polytypic polycrystalline perovskite results in ultra-long carrier lifetimes exceeding 18 µs, and affords PSCs with a PCE of 24.13%, a module efficiency of 22.83% (active area: 23.2 cm²) and long-term stability.

## Results
### Configuration of hetero-polytypic perovskite
The formation of the 6H polytype was achieved by the addition of 20% excess PbI$_2$ (0.24 M) based on FAPbI$_3$ (1.2 M), combined with 55 mol% methylammonium chloride (MACl, 0.66 M). The control film was obtained from the same precursor but without excess PbI$_2$. Compared to previous studies, a higher concentration of PbI$_2$ and MACl was used in order to form the 6H polytype perovskite (Supplementary Note 1 and Supplementary Tables 1 and 2)[25]. The conditional formation of the 6H phase was demonstrated through a comparison of the films with varying concentrations of PbI$_2$ and MACl, respectively (Supplementary Figs. 1 and 2), and we propose a plausible underlying mechanism for the formation of the 6H polytype (Supplementary Note 2 and Supplementary Figs. 3–6). Besides, we investigated whether PbCl$_2$ could replace excess PbI$_2$ in forming the 6H phase, resulting in an extremely weak diffraction signal of 3 C (Supplementary Fig. 7).

X-ray diffraction (XRD) of the films exhibits a typical diffraction profile for the cubic polytype (3 C) FAPbI$_3$, with the main peaks at 14.3 and 28.4°, corresponding to (100) and (200) (Fig. 1a). Figure 1b shows the XRD patterns in the low $2\theta$ range between 10 and 13.6° with a prominent diffraction peak at 13.0° corresponding to unreacted PbI$_2$ observed only in the film containing excess PbI$_2$[26]. In the control film, a diffraction peak is observed at 12.1° that originates from face-sharing hexagonal polytype 2H-FAPbI$_3$, also known to be an undesirable $\delta$ phase (non-perovskite), which may be attributed to the phase transition from the $\alpha$ phase by the exposure of the film to air containing moisture[3,27]. In contrast, the addition of excess PbI$_2$ suppresses the transformation of FAPbI$_3$ from the $\alpha$ phase to the $\delta$ phase, due to the formation of the 6H perovskite that is more likely to be formed along with the formation of the 3 C phase exhibiting (101) diffraction observed at 12.5°. To understand the role of the 6H phase in stabilising the 3 C phase, the thermodynamics of the 2H, 6H, and 3 C phases were computationally evaluated by performing Born-Oppenheimer molecular dynamics simulations at specific temperatures of 300 and 423 K (Supplementary Note 3 and Supplementary Fig. 8). As the 6H phase is more thermodynamically stable than the 2H phase, it creates a kinetic barrier for the interconversion of the 3 C phase to the 2H phase via the 6H phase.

In addition, the use of excess PbI$_2$, coupled with the high concentration of MACl, results in films that display weak diffraction peaks at 15.7, 31.0 and 31.5°, which may be attributed to (100) and (200) of Cl-based perovskite (e.g. MAPbCl$_3$ or FA, I-doped MAPbCl$_3$) (Supplementary Fig. 6)[28]. It is difficult to quantify the amount of the Cl-based perovskite incorporated in the FAPbI$_3$ matrix since the amount of Cl$^-$ contributing to the anion exchange process is not known[25]. However, it is conceivable that its amount in the film is very small as the phase yield of the Cl compound appears to be 200 times lower than that of FAPbI$_3$ based on relative diffraction intensities of 0.5% (Supplementary Fig. 11).

To corroborate the existence of different polytypes in the films beyond the appearance of thin film diffraction peaks, single crystals of the perovskites were synthesized by a liquid-liquid diffusion method using the same constituents and composition employed for the film fabrication[29]. Single crystal X-ray diffraction (SCXRD) confirmed the presence of the hexagonal phases, i.e. 2H, 6H, as well as of the cubic phase 3 C (Supplementary Table 6). The aristotype 3 C perovskite adopts a cubic close-packed (c) structure that results in a corner-sharing PbI$_6$ octahedra framework. In the 2H perovskite, the close-packing is hexagonal (h), rearranging octahedra to form 1D face-sharing chains (Supplementary Fig. 12). The anisotropic nature of the 1D structure of 2H-FAPbI$_3$ disfavors efficient carrier transport in PSCs because electron hopping is mainly along the 1D chains, and inter-chain charge transfer is more challenging due to the intercalated organic molecule[30]. In contrast, the 6H polytype, which is a combination of both h and c stacking sequences in the order of cchcch… allows charge carriers to have a higher degree of freedom in their hopping in the bulk with suppressed quenching, because the corner-sharing halides multi-directionally connect the octahedral carrier-hopping channels (Supplementary Fig. 12).

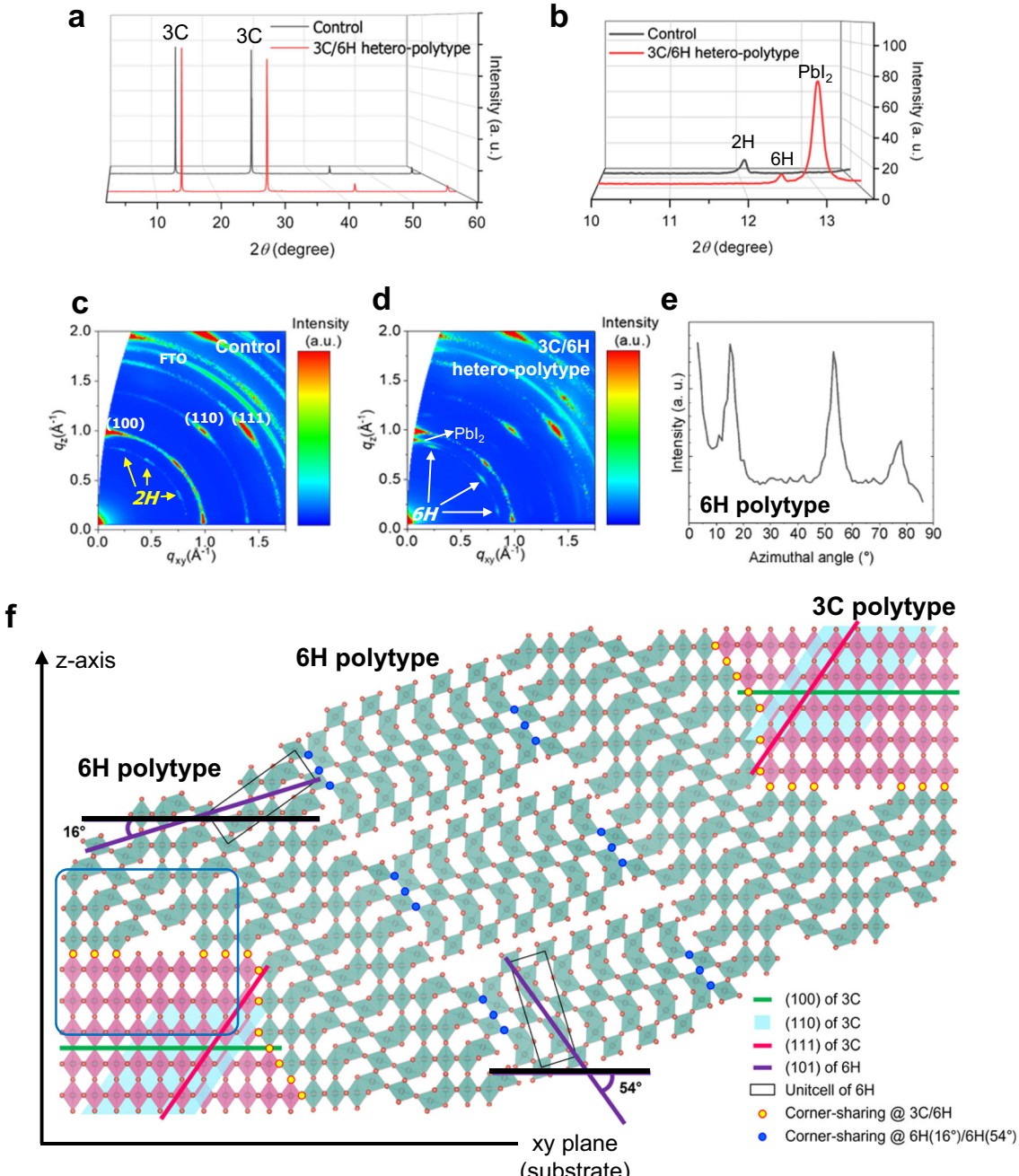

**Fig. 1 | Configuration of the hetero-polytypic perovskite compared to the control film. a** X-ray diffraction (XRD) patterns of the perovskite films. **b** Magnified XRD patterns of the films in the low $2\theta$ angle region. **c**, **d** 2D grazing-incidence wide-angle X-ray scattering (GIWAXS) pattern of the control film (**c**) and the 3 C/6H hetero-polytypic perovskite film (**d**). **e** Azimuthally integrated diffraction intensity of 6H polytype within the hetero-polytypic perovskite film. **f** A lattice map exhibiting the configuration of hetero-polytypic perovskite composed of 3 C and 6H phases with high interphase coherency. The visualization was accomplished using VESTA[31].

Further information about the crystalline nature of the perovskite films was obtained using grazing-incidence wide-angle X-ray scattering (GIWAXS). Figure 1c and d show the diffraction images of the bulk films with an incident angle of 0.40°. Indexing of Bragg spots of the 3 C polytype, (100), (110) and (111) indicates a highly ordered material with the (100) being oriented out-of-plane, which is corroborated by the azimuthal angle-dependent diffraction intensity of the planes (Supplementary Fig. 13). GIWAXS also confirmed the presence of the hexagonal perovskite polytypes; the film with excess $PbI_2$ exhibits the (101) diffraction of the 6H polytype at $q = 0.856\,\text{Å}^{-1}$, and the diffraction by the 2H polytype was observed in the control film at $q = 0.825\,\text{Å}^{-1}$, which

agrees with the observation by the regular 1D Bragg Brentano diffraction shown in Fig. 1b. Importantly, the orientation of the hexagonal polytypes reveals a strong directional dependency on the orientation of the 3 C phase. The strong directional dependency implies structural integrity of the grains, possibly based on a coherent interface between the hexagonal polytypes and the 3 C polytype. The azimuthal integration of the (101) diffraction of the 6H polytype exhibited intensity maxima at 14-17° and 52-55° (Fig. 1e). The increasing intensity below 9° may be attributed to the intensity overlap with out-of-plane $PbI_2$.

Based on the abovementioned XRD results, a lattice map of the perovskites consisting of the 3 C and 6H polytypes was deduced

(Fig. 1f)[31]. The 6H polytype with a tilt angle of 16° aligns well with the out-of-plane 3 C polytype, showing high coherency along with the intervening halides (indicated as yellow circles). The 6H polytype with a tilt angle of 54° can coherently bridge the 16° tilted 6H phase on the other side via corner-sharing (with the other corner – indicated as blue circles) so that the separate 3 C phases are ultimately linked. 3D lattice arrays of the 3 C (100)−6H (101) interface (in the blue boxed area in Fig. 1f) are presented in various directions (Supplementary Fig. 14). As several facets of the 6H-FAPbI$_3$ phase, including (110) demonstrate an iodine-rich configuration compatible with the topology of (110) surface of 3C-FAPbI$_3$ which is one of the most abundant surfaces, we accordingly performed molecular dynamics simulations of the 3 C/6H interface which indicate that the 6H-FAPbI$_3$ can heal the iodine vacancies on the surface of the 3C-FAPbI$_3$ by filling them (Supplementary Movie 1). The high lattice coherency at the interface of 3 C/6H in this configuration presumably minimizes non-radiative recombination loss as the density of defect at the intervening sites reduces. Moreover, the improved structural integrity could increase the moisture resistance of the polycrystalline film as diffusion of water along the incoherent interface is impeded by the intervening iodides[32].

Scanning electron microscopy (SEM) distinguished the surface morphology of the perovskite films as the contrast in the resulting image reflects the compositional information with the intensity of backscattered electrons, which is dependent on the average atomic number[33]. (Supplementary Fig. 15). As PbI$_2$ has a higher average atomic number than the FAPbI$_3$ perovskite, it is brighter in the SEM image. Notably, the PbI$_2$ grains were only at the upper part of the perovskite layer when observed from its cross-section, and the incorporation of the 6H polytype led to a more than doubled average grain size with an increase in film thickness compared to the control film (Supplementary Fig. 16, Supplementary Table 7 and Supplementary Note 4). The underlying reason for the increase in grain size and film thickness may be attributed to the presence of the 6H phase in the film. As the 6H polytype has a large number of corner-sharing PbI$_6$ octahedra that can act as inter-grain growth channels by the intervening halides, grain growth can be facilitated by a high degree of lattice expandability. In contrast, the grain growth without the 6H polytype is limited by the disconnected 1D face-sharing octahedra strings of the 2H polytype. The reduced density of grain boundaries within the 3 C/6H hetero-polytypic perovskite film due to the increase in the grain size may lead to improved photo-physical properties.

## Charge carrier dynamics in hetero-polytypic perovskite

The steady-state PL intensity of the 3 C/6H hetero-polytypic perovskite film increases dramatically at ca. 810 nm compared to the control film (Fig. 2a). Also, the transient PL measurement of the hetero-polytypic perovskite film exhibited a considerable increase in PL lifetime; the film with the 6H phase has a PL average lifetime ($\tau_{Ave}$) of 6.73 μs whereas the control film has a $\tau_{Ave}$ of 0.27 μs (Fig. 2b and Supplementary Table 8). The significant improvement in PL implies that the incorporation of 6H polytype into polycrystalline perovskite effectively suppresses non-radiative recombination.

To understand the improved carrier dynamics in the 3 C/6H hetero-polytypic perovskite system, we scrutinized the impact of the major defect $V_I^+$ that otherwise remains at the grain boundary of the 3 C phase without the 6H polytype. The modeling of the formation of $V_I$ with different valencies ($q$ = +1, 0, −1) revealed that $V_I^+$ is the dominant defect in the FAPbI$_3$ interface due to its lowest formation energy (Supplementary Note 5). The electronic structure calculations for the (110) surface of 3C-FAPbI$_3$ with the presence of $V_I^+$ demonstrate the formation of shallow trap states accompanied by the formation of moderately localized hole polaron that may be benign and not significantly affect the radiative recombination dynamics (Fig. 2c and d). Considering charge transition levels (CTLs) of the defect, however, changes the situation.

Computation of the energy of the CTLs of $V_I$ reveals that the transition +1/0 is energetically the easiest and is situated below the conduction band edge (Supplementary Note 6). Thus, the moderately localized hole polaron can easily trap an excited electron in the conduction band to reduce $V_I^+$ to $V_I^0$ and it transforms into a small electron polaron localized at the vacancy site along with the formation of deep trap states that can in turn lead to non-radiative recombination of charge carriers (Fig. 2e and f). Thus, the employment of the 6H polytype in the hetero-polytypic perovskite to screen $V_I^+$ is pivotal to preventing electron trapping and the formation of the localized small polaron that accompanies deep-level trap states. We present a comprehensive schematic of the carrier dynamics occurring in the presence of $V_I^+$ in Fig. 2g.

Absorption spectra also reveal differences due to the incorporation of the 6H polytype (Fig. 3a). For example, in the 6H bridged polycrystalline film the secondary onset is at 550 nm (Supplementary Fig. 17), which is indicative of the presence of the polytype 6H[29]. Moreover, an intriguing difference between the films with respect to the slope of the absorption bands of the 3 C phase is observed, i.e. the 3 C/6H hetero-polytypic perovskite film exhibits a more gradual slope compared to the control film. Elliott's theory was implemented to characterize this feature (Supplementary Note 7). The deconvolution of the absorption spectra distinguished the contribution from the excitonic resonance and from the continuum state, overall revealing a significant change in the optoelectronic behavior of the film with the 6H polytype compared to the control film (Figs. 3b and 3c). Compared to the control film, in the film with the 6H polytype the value of $E_b$ decreases from 27.88 to 19.19 meV and the linewidth of the excitonic resonance ($\Gamma_{ex}$) narrows from 27.39 to 22.08 meV (Fig. 3d). As the presence of $V_I^0$ is responsible for the formation of deep trap states, and the point defect can be accumulated at the extended defects such as grain boundary and interphase boundary[34,35], it is likely that the phase boundary is occupied by the localized electron small polarons. In this sense, the reduction of $E_b$ may be attributed to the mitigation of the local electron polaron states due to defect screening by the 6H polytype bridges that neutralize the charged point defect sites with the intervening halides[36,37]. In contrast, in the absence of the 6H bridges, the incoherent phase boundary facilitates non-radiative recombination due to the localized dielectric environment along the interface[38]. Besides, the decrease in $\Gamma_{ex}$ of the film with the 6H polytype can be correlated with reduced phonon scattering due to the defect screening, whereas the spectral-broadening of the excitonic resonance of the control film can arise from the strong electron-phonon coupling that is highly impacted by the formation of localized polarons.

To further study electron-phonon coupling, we performed temperature-dependent PL measurements of the films and extracted relevant parameters, i.e. temperature-independent inhomogeneous PL broadening ($\Gamma_0$), representative energy of longitudinal optical (LO) phonon ($E_{LO}$), and electron-LO phonon coupling strength ($\gamma_{LO}$) (Fig. 3e and f, and Supplementary Fig. 18), all of which decrease following the incorporation of the 6H polytype (Fig. 3g and Supplementary Table 9). The reduction of $\Gamma_0$ indicates reduced disorder in the lattice by incorporating the 6H phase, and thus the less intense electron-LO phonon coupling results in lower $\gamma_{LO}$ and $E_{LO}$ values. This can be understood in terms of the effective defect screening along the interface between the 3 C phases by the intercalated 6H polytype, which makes the local polaronic environment less intense. In other words, the passivation of the defects at the surface of 3C-FAPbI$_3$ reduces the number of dangling bonds and increases the effective mass participating in the vibrations, naturally shifting the phonon spectra at the interphase to lower values and suppressing the higher frequency phonons[39,40], which are typically more efficient in radiative recombination processes (Fig. 3h). The difference in the extent of the electron-phonon coupling can, in turn, affect charge carrier mobility. The analysis of the space-charge-limited current of monopolar devices

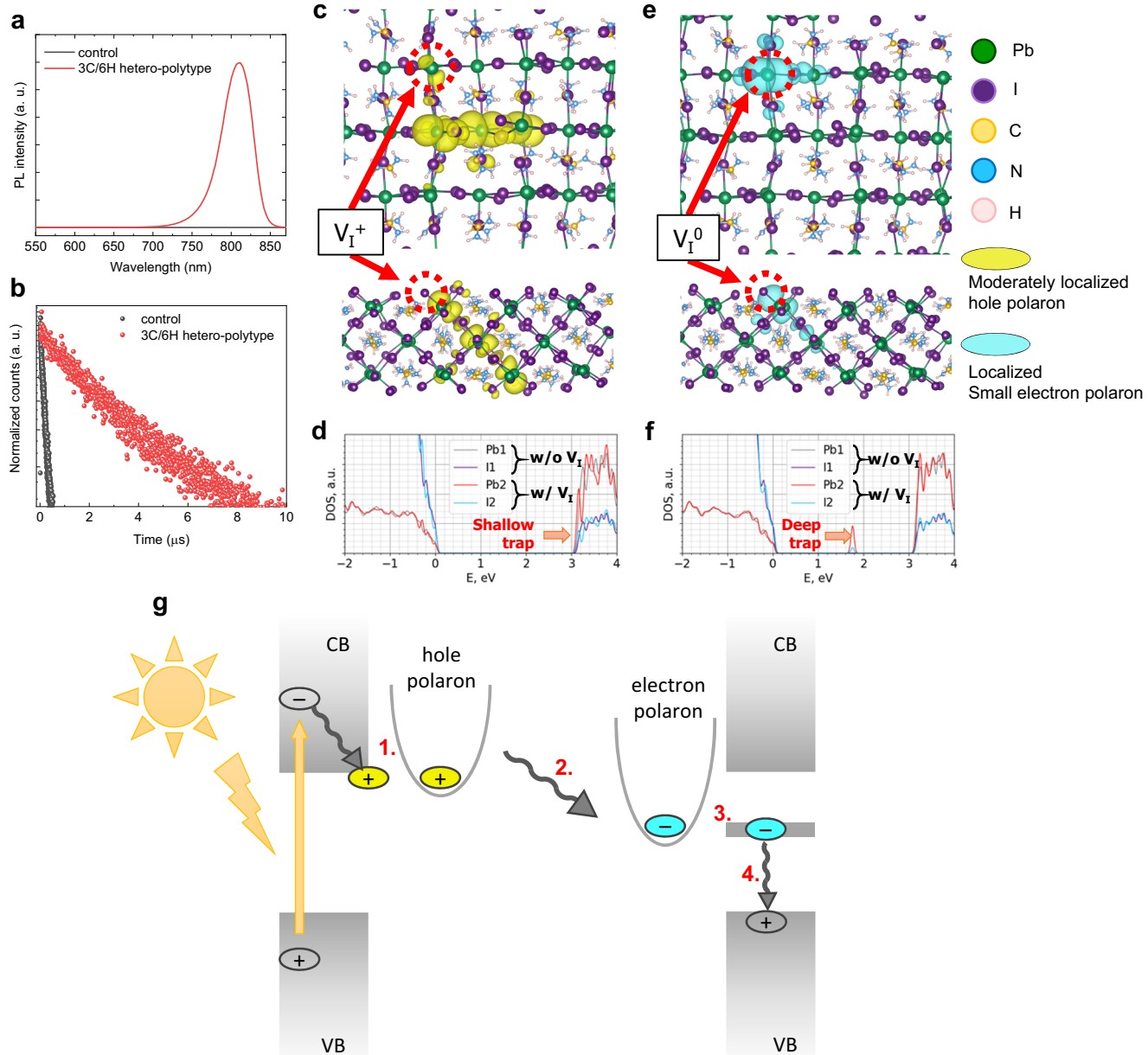

**Fig. 2 | Detrimentality of shallow-level defect $V_I^+$. a, b** Steady-state photo-luminescence (PL) (**a**) and transient PL (**b**) of the control film and the 3 C/6H hetero-polytypic perovskite film. **c, d** Top (top) and side (bottom) view (**c**) of (110) surface of 3C-FAPbI$_3$ in the presence of $V_I^+$ showing the formation of moderately localized hole polaron that demonstrates the formation of shallow trap states in the density of states (**d**). **e, f** Top (top) and side (bottom) view (**e**) of (110) surface of 3C-FAPbI$_3$ in the presence of $V_I^0$ by the reduction of $V_I^+$ and the formation of

localized small electron polaron that demonstrates the formation of deep trap states in the density of states (**f**). **g** Schematic of the carrier dynamics in the presence of $V_I^+$: 1. $V_I^+$ generates hole states at the bottom of the CB, and induces the formation of hole polarons. 2. An electron is trapped at the $V_I$ site to reduce $V_I^+$ into $V_I^0$, 3. that leads to the formation of small electron polarons and deep-level trap states. 4. This facilitates non-radiative recombination.

confirms an increase in $\mu_e$ of the 3 C/6H hetero-polytypic perovskite film compared to the control film (Supplementary Fig. 19). This increase may be attributed to the reduced electron-phonon coupling, and also to suppressed non-radiative recombination by the 6H poly-type that may act as a bridge to facilitate carrier transport (Fig. 3i). In contrast, carriers in the control film without the 6H bridge are quen-ched by polaronic barriers. Also, the use of the 6H polytype reduces the trap density ($n_t$) of the perovskite film (Supplementary Fig. 19). Interestingly, we found out that the reduction of trap density while considering the increase in grain size corresponds to the theoretical prediction of carrier lifetime extension in the 3 C/6H hetero-polytypic perovskite (Supplementary Note 8).

## Surface passivation by low-dimensional perovskite
As mentioned in the introduction, surface passivation has become a standard procedure to enhance the efficiency and stability of PSCs, by reducing the density of defects at the surface, increasing moisture resistance, and stabilizing the interface between the absorption layer and the hole transporting layer, spiro-OMeTAD[17,41]. Thus, the addi-tional defect passivation involving deposition of a surface passiva-tion layer of octylammonium iodide (OAI) on top of the 3 C/6H hetero-polytypic perovskite film was performed to induce the for-mation of a LDP passivation layer[42,43]. GIWAXS confirms that the deposition leads to the formation of LDP indicated by a new dif-fraction pattern at a scattering vector of $q \sim 0.24\,\text{Å}^{-1}$ (Fig. 4a and

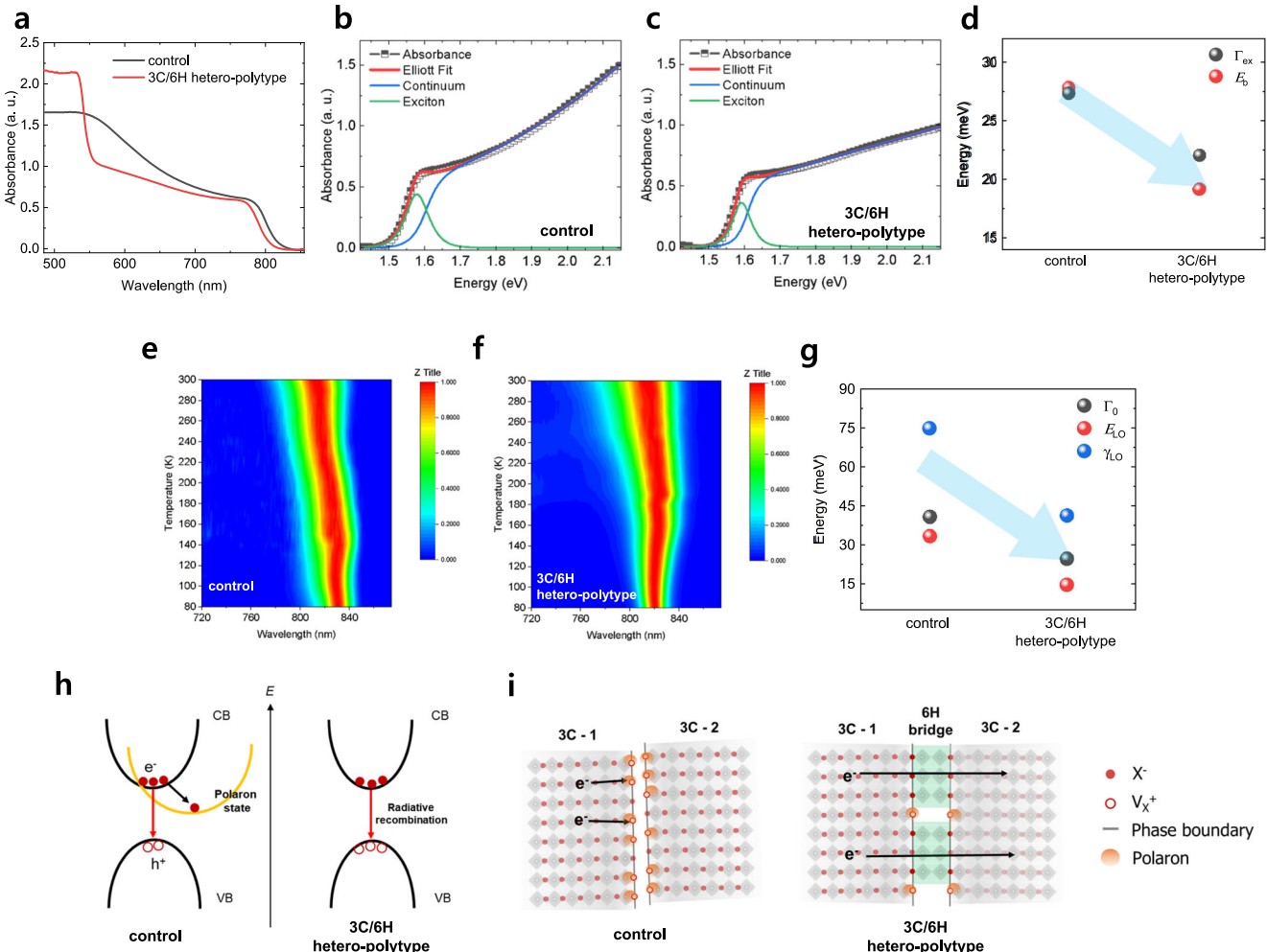

**Fig. 3 | Charge carrier dynamics in the hetero-polytypic perovskite. a** UV-visible absorption spectra of the control film and the 3 C/6H hetero-polytypic perovskite film. **b**, **c**, Elliott fits to the absorption spectra of the control film (**b**) and the 3 C/6H hetero-polytypic perovskite film (**c**). **d** Reduction of linewidth of the excitonic resonance ($\Gamma_{ex}$) and exciton binding energy ($E_b$) of the perovskite with the incorporation of the 6H polytype. **e**, **f** Normalized temperature-dependent steady-state PL of the control film (**e**) and the 3 C/6H hetero-polytypic perovskite film (**f**).

**g** Reduction of temperature-independent inhomogeneous PL broadening ($\Gamma_0$), energy of longitudinal optical (LO) phonons ($E_{LO}$), and coupling strength of electron-LO phonon ($\gamma_{LO}$) of the perovskite with the incorporation of the 6H polytype. **h** Schematic of carrier recombination dynamics of the control perovskite and the hetero-polytypic perovskite. **i**, Schematic of carrier transport in perovskite without 6H (left) and with 6H bridge (right) that interconnects two separate 3 C phases.

Supplementary Fig. 20). The underlying 3D perovskite remains intact with the 6H bridge phase retaining its structural integrity after the deposition of OAI (Fig. 4b). The surface morphology of the film after OAI-deposition shows that the brighter $PbI_2$ grains were converted into the dark phase with less contrast, indicative of the formation of perovskite (Supplementary Fig. 21). From the cross-section of the film, the bright phase of the unreacted $PbI_2$ beneath the LDP and in between grains is observed, which can contribute to local passivation of defects, particularly in the upper part of the perovskite layer (Fig. 4c). The formation of the LDP results in additional PL peaks at 568 and 514 nm, corresponding to $n = 2$ and $n = 1$ phases, respectively (Supplementary Fig. 22). Notably, the introduction of the 2D perovskite on top of the hetero-polytypic 3D perovskite results in a considerably prolonged carrier lifetime ($\tau_{Ave} = 18.95$ μs) (Fig. 4d and Supplementary Table 8).

### Evaluation of perovskite solar cells and modules
The films were evaluated in PSCs with an n-i-p structure (FTO/compact-TiO$_2$/mesoporous TiO$_2$/KCl/perovskite/spiro-OMeTAD/Au) (Fig. 5a and b, Supplementary Fig. 23, and Supplementary Table 10). The control device gave photovoltaic parameters corresponding to $V_{oc}$ of

1.098 V, $J_{sc}$ of 24.40 mA cm$^{-2}$, FF of 75.82%, and a PCE of 20.32%. The 6H bridged hetero-polytypic polycrystalline film resulted in significantly improved photovoltaic characteristics, i.e. $V_{oc}$ of 1.145 V, $J_{sc}$ of 24.80 mA cm$^{-2}$, FF of 78.72%, and a PCE of 22.35% in the best cell. A further improvement in device efficiency was achieved by the introduction of the additional LDP passivation layer on top of the 6H bridged perovskite film. The champion device resulted in a $V_{oc}$ of 1.156 V, $J_{sc}$ of 25.58 mA cm$^{-2}$, FF of 81.60%, and a PCE of 24.13%. One of the best-performing devices was certified by the Korea Institute of Energy Research to achieve a PCE of 23.69% (Supplementary Fig. 24). The steady-state efficiency of the devices was measured by tracking the maximum power point (MPP). The stabilised efficiencies of the devices were 19.43, 21.72, and 23.93%, respectively (Supplementary Fig. 25). Accordingly, the external quantum efficiency (EQE) of the devices showed the same trend, and the integrated $J_{sc}$ calculated from the EQE well matched the $J_{sc}$ from the J-V curves (Supplementary Fig. 26). Investigation of the ideality factor ($n_{id}$) from light-intensity dependent $V_{oc}$ corroborates the superiority of the 6H bridged perovskite and the surface passivation in terms of carrier dynamics (Supplementary Fig. 27). Perovskite solar modules were fabricated using the 3 C/6H hetero-polytypic perovskite with LDP passivation on a

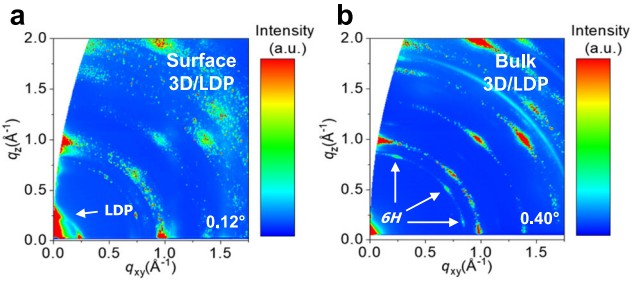

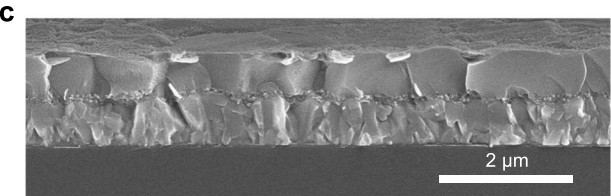

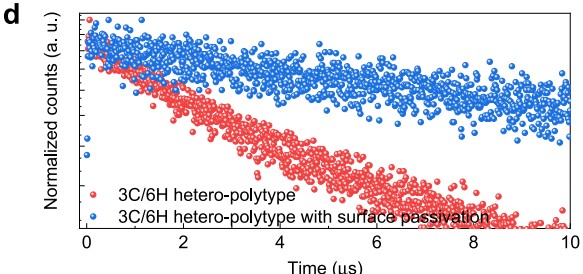

**Fig. 4 | Surface passivation by the low-dimensional perovskite. a, b** GIWAXS patterns of the 3 C/6H hetero-polytypic perovskite with surface passivation with X-ray incident angles of 0.12° (**a**) and 0.40° (**b**) to analyze the surface and bulk of the film, respectively. **c** Cross-sectional SEM image of the hetero-polytypic perovskite with surface passivation. **d** Transient PL decay of the hetero-polytypic perovskite without and with surface passivation.

$6.5 \times 7.0$ cm² substrate with 8 sub-cells connected in series by P1-P2-P3 interconnection. The best performing device showed a PCE of 21.92% with a $V_{oc}$ of 9.227 V, $J_{sc}$ of 3.07 mA cm⁻² and FF of 77.4% under a shadow mask having an aperture area of 28.62 cm², which corresponds to an active PCE of 22.57% given the geometric fill factor (gFF) of 97.1%. (Fig. 5c). The module was certified at Newport Photovoltaic Testing and Calibration Laboratory in the US showing a certified PCE of 21.44% with a $V_{oc}$ of 9.03 V, $J_{sc}$ of 3.11 mA cm⁻² and FF of 76.32% under reverse bias (Supplementary Fig. 28 and Supplementary Fig. 29), which corresponds to an active PCE of 22.08% given the gFF. Note that the $J$-$V$ hysteresis observed in the devices and modules can be attributed to the presence of residual PbI₂ in the films (Supplementary Fig. 15), indicating that further enhancement in device performance could be achieved by optimising the amount of residual PbI₂ on the hetero-polytypic perovskite.

Furthermore, we demonstrated low-temperature processed modules with a structure of ITO/SnO₂/KCl/hetero-polytypic perovskite/LDP/spiro-OMeTAD/Au on a $7.0 \times 7.0$ cm² substrate consisting of 10 sub-cells in series by P1-P2-P3 interconnection (Supplementary Fig. 30). The highest PCE observed for the modules was 21.74%, with a $V_{oc}$ of 12.18 V, $J_{sc}$ of 2.23 mA cm⁻² and FF of 79.96% on an aperture area of 24.5 cm², which corresponds to an active PCE of 22.96% given the gFF of 94.7% (Fig. 5d). The average PCE of 30 modules was 21.20%, with a standard deviation of 0.25, indicating good reproducibility (Supplementary Fig. 31 and Supplementary Table 11). PL imaging of the large-area perovskite films of the low-temperature processed module confirmed the superiority of the hetero-polytypic perovskite in terms of radiative recombination compared to the control film

(Supplementary Fig. 32). For comparison, further details concerning the devices and modules are provided in Supplementary Table 12. Additionally, an overview of the performance of recently reported perovskite solar modules, including our results are summarized in Supplementary Table 13 and Supplementary Fig. 33 for comparison purposes.

### Stability of hetero-polytypic perovskite solar cells

The 6H bridge also contributes to an improvement in the stability of the film and consequently to the PSC device. The films were stored at room temperature under a relative humidity of > 90% (Supplementary Fig. 34). The control film without the 6H phase completely decomposed after one day under these conditions. In contrast, the film with the 6H phase maintained its black phase for more than two days, demonstrating its higher moisture resistance. This enhanced stability is attributed to the high coherency between the 3 C and 6H phases, which likely impedes the diffusion of water. The additional surface passivation further improved the stability of the film allowing the black phase to last for more than five days due to the hydrophobicity of the LDP coating. Enhanced moisture resistance is supported by the water contact angle measurements of the films (Supplementary Fig. 35). MPP tracking of the devices under constant 1 sun illumination under an inert atmosphere further demonstrated the excellent stability imparted by the 6H bridged perovskite and the LDP layer. The control device retained 63% of its initial PCE after 1000 h whereas the device containing the 6H polytype perovskite retained > 81% and that with the LDP layer maintained > 92% (Fig. 5e).

## Discussion

We report the use of a hexagonal polytype (6H), with its corner-sharing component in the $\alpha$-phase cubic polytype 3C-FAPbI₃ to construct a 3 C/6H hetero-polytypic perovskite that may be close to the ideal configuration of a polycrystalline perovskite film as it maintains material homogeneity and integrity. The corner-sharing constituent of the 6H polytype stabilizes the lattice at the surface of the 3C-FAPbI₃ with high coherency to screen the major defect $V_I^+$ at the boundary. Importantly, based on both theoretical and experimental results, we reveal a complex non-radiative recombination mechanism initiated by the dominant shallow-level defect $V_I^+$ that eventually evolves to deep-level defect $V_I^0$. Thus, defect screening by the 6H polytype leads to a significant improvement in carrier dynamics of the hetero-polytypic 3 C/6H perovskite. Moreover, the high lattice coherency at the 3 C/6H interface contributes to the enhancement of film stability by impeding water diffusion. Additional LDP passivation on top of the hetero-polytypic perovskite film results in an ultra-long carrier lifetime exceeding 18 µs. We achieve PSCs with a PCE of 24.13% and long-term stability. Also, we demonstrate a perovskite solar module with a PCE of 21.92% (certified PCE: 21.44%) and a low-temperature processed module with a PCE of 21.74%. We believe that our findings pave the way for the employment of perovskite polytypes in practice to achieve high-performing PSCs concurrently elucidating the necessity of engineering shallow-level defects as they strongly impact carrier recombination dynamics.

## Methods

### Materials

All chemicals were used without further purification. Formamidinium iodide (FAI), lead iodide (PbI₂), methylammonium chloride (MACl), octylammonium iodide (OAI), 4-tert-butylpyridine (tBP) and 2,2',7,7'-tetraakis-(N,N-di-4-methoxyphenylamino)−9,9'-spiro-bifluorene (Spiro-OMeTAD, 99.8%) were purchased from Xi'an polymer light technology corp. Dimethylformamide (DMF, 99.8%), dimethyl sulfoxide (DMSO, 99.8%), isopropanol (IPA, 99.8%), acetonitrile (99.9%), and diethyl ether (99.5%) were purchased from Acros Organics. A

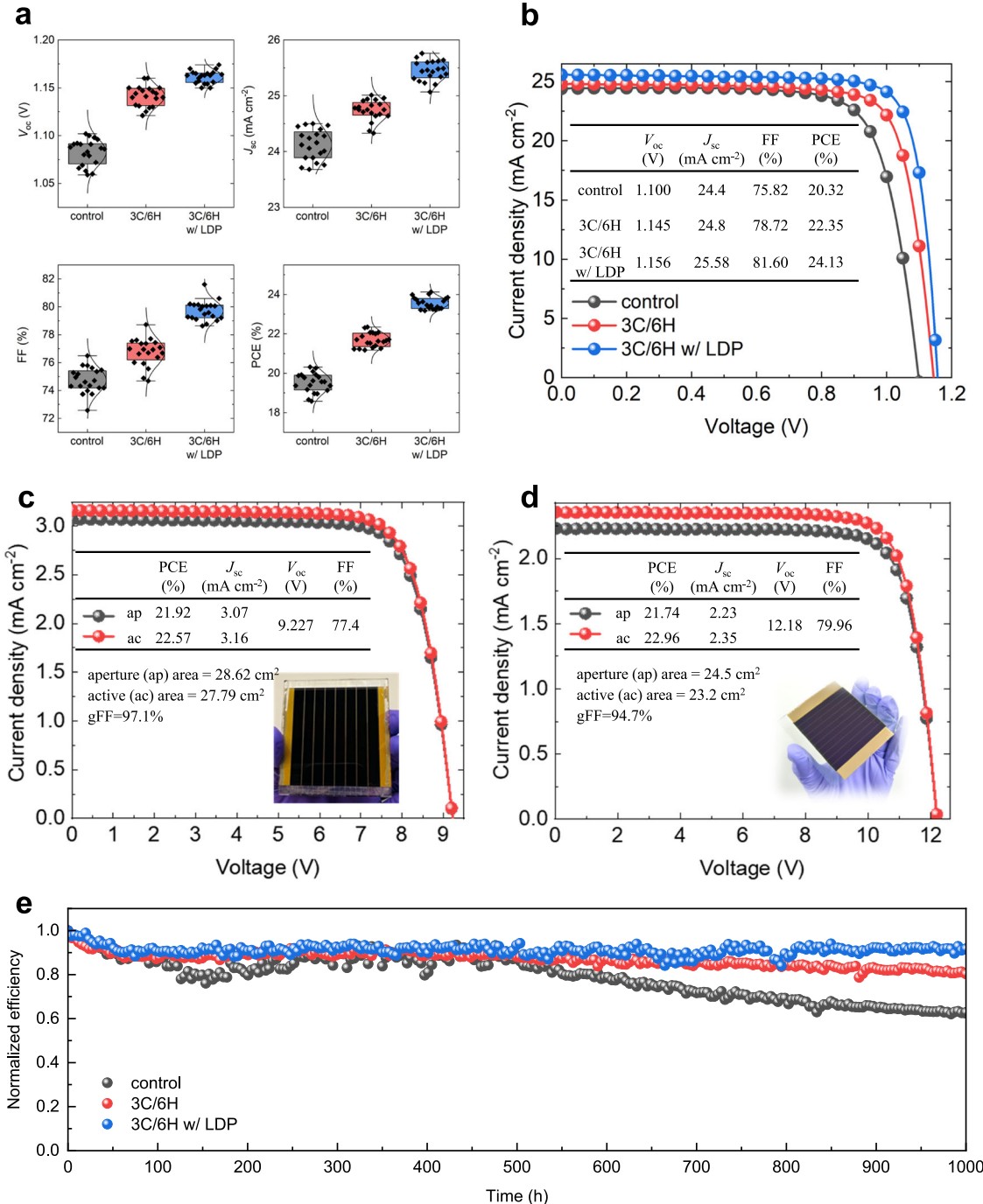

**Fig. 5 | Perovskite solar cell and module performance. a** Statistics of photovoltaic performance ($V_{oc}$, $J_{sc}$, FF and PCE) of perovskite solar cells (PSCs). **b** $J$-$V$ curves of the PSCs using the control perovskite (black), the 3 C/6H hetero-polytypic perovskite (red), and the hetero-polytypic perovskite with surface passivation (blue) under reverse bias. **c** $J$-$V$ curves of the perovskite solar module using the hetero-polytypic perovskite with surface passivation under reverse bias. **d** $J$-$V$ curves of the low-temperature processed perovskite solar module using the hetero-polytypic perovskite with surface passivation under reverse bias. **e** Long-term stability of the PSCs under constant 1 sun illumination.

mesoporous $TiO_2$ paste (SC-HT040) was purchased from Sharechem. Titanium diisopropoxide bis(acetylacetonate), tin(IV) bis(acetylaceto-nate) dichloride ($Sn(acac)_2Cl_2$, 98%), bis(trifluoromethane)sulfonimide lithium salt (Li-TFSI), potassium chloride (KCl), lithium carbonate ($Li_2CO_3$) and chlorobenzene (99.8%) were obtained from Sigma-Aldrich. Tin(IV) oxide ($SnO_2$, 15% in $H_2O$ colloidal dispersion) was purchased from Alfa Aesar. Cobalt(III) FK209 TFSI Salt was purchased from Greatcell Solar and ethanol (99.8%) was obtained from Fisher Scientific.

## Fabrication of perovskite solar cells

FTO glass substrates (Asahi FTO glass) were sequentially cleaned with detergent, deionized (DI) water, acetone, and IPA. A compact $TiO_2$ layer was deposited by spray pyrolysis method on the cleaned FTO substrate heated at 450 °C. The precursor solution was prepared by diluting titanium diisopropoxide bis(acetylacetonate) in IPA (1:15, v/v). A mesoporous $TiO_2$ layer was spin coated using diluted $TiO_2$ paste in ethanol (0.112 g mL$^{-1}$) at 1500 rpm for 40 s, followed by sin-tering at 500 °C for 30 min. 30 mM of KCl aqueous solution was spin

coated at 3000 rpm for 30 s on top of the mesoporous $TiO_2$ layer followed by heating at 100 °C for 10 min. In order to deposit perovskite, the substrates were transferred into a glovebox after UV-ozone treatment for 15 min. The stoichiometric perovskite solution was prepared by dissolving 206.4 mg of FAI (1.2 M), 553.2 mg of $PbI_2$ (1.2 M), and 44.6 mg of MACl (0.66 M) in a mixed solvent of DMF:DMSO (8:1, v/v). The precursor solution with excess $PbI_2$ was prepared by mixing 206.4 mg of FAI (1.2 M), 663.9 mg of $PbI_2$ (1.44 M), and 44.6 mg of MACl (0.66 M) in a mixed solvent of DMF:DMSO (8:1, v/v). The perovskite solution was spin coated on the as-prepared substrate at 1000 rpm for 10 s followed by 6000 rpm for 30 s. During the second step of spin coating, 500 μL of diethyl ether was deposited on the spinning substrate. The deposited film was annealed at 150 °C for 13 min. For surface passivation, 100 μL of OAI solution (10 mg mL$^{-1}$ in isopropanol) was spin coated on the perovskite layer at 4000 rpm for 20 s followed by thermal annealing at 100 °C for 5 min. After cooling the substrate, spiro-OMeTAD solution, prepared by mixing with 80 mg of spiro-OMeTAD in 1023 μL of chlorobenzene, 32 μL of tBP, 19 μL of Li-TFSI solution (517 mg mL$^{-1}$ in acetonitrile), and 14 μL of Co-TFSI solution (376 mg mL$^{-1}$ in acetonitrile), was spin coated on the perovskite layer at 4000 rpm for 20 s. Finally, a 70 nm-thick gold layer was thermally evaporated to complete the device.

### Fabrication of perovskite solar modules

FTO glass substrates with a size of 6.5 cm ×7.0 cm were patterned with 8 sub-cells connected in series. The series interconnection of the module was realized by P1, P2, and P3 lines, which were patterned using a laser scribing system with 1064 nm and a power of 20 W (Trotec). The FTO substrate was pre-patterned for P1 (width of 30 μm) by means of 80% laser power under a speed of 300 mm/s with a frequency of 65 kHz and pulse width of 120 ns. The subsequent processes for the preparation of ETL layer, perovskite layer, and Spiro-OMeTAD layer are the same as the small-area device procedures. The P2 lines (width of 60 μm) were patterned before the Au evaporation process step with an average laser power of 15% under a speed of 1000 mm/s and frequency of 65 kHz for a pulse duration of 120 ns. After Au layer were sequentially deposited, the P3 line (width of 30 μm) was fabricated under the same scribing condition as the P2 line. The width of dead area is about 190 μm, and a geometric fill factor (gFF) is around 97.1%.

For the low-temperature processed modules, ITO glass substrates (7 cm × 7 cm) were patterned by a laser to form 10 series connected cells with a power of 1.38 W (P1). A $Sn(acac)_2Cl_2$ solution dissolved in 2-methoxyethanol/IPA (1:1) was shearing-coated on the substrate using a multi coater (PMC-300, PEMS Korea) at a speed of 5 mm s$^{-1}$ with a gap of 100 μm, and then annealed at 160 °C for 1 h. Subsequently, $SnO_2$ was shearing-coated with 100 μL of precursor solution on the multi coater at a speed of 0.5 mm s$^{-1}$ with a gap of 100 μm. The film was annealed at 100 °C for 30 min. Lithium carbonate aqueous solution (1 mg mL$^{-1}$) was treated on the substrates by spin-coating method and then annealed at 100 °C for 30 min. Perovskite precursor solution was prepared by dissolving 800 mg of $FAPbI_3$, 30 mg of $MAPbBr_3$, 30 mg of MACl, and 116 mg of $PbI_2$ in 1 mL of DMF/DMSO mixed solvent (85:15). Perovskite film fabrication was conducted by dispensing 100 μL of perovskite precursor solution on the multi coater and shearing coating at a speed of 5 mm s$^{-1}$ with a gap of 100 μm. The printed film was dried in the air, and then immediately immersed in a bath of mixed antisolvent (tert-butyl alcohol:ethyl acetate=7:3, v/v) for 20 s. Subsequently, the film was blown with compressed air, and annealed at 100 °C for 1 h. For surface passivation, OAI solution was spin-coated on the perovskite layer, and annealed at 100 °C for 5 min. A spiro-OMeTAD solution, prepared by the same procedures described above, was shearing-coated on the as-prepared substrate at a speed of 12 mm s$^{-1}$ with a gap of 100 μm. P2 laser scribing was conducted with a power of 1.01 W, and

then Au counter electrode was deposited by thermal evaporation. P3 laser scribing was processed with a power of 0.12 W. The calculated gFF was 94.7%.

### Photovoltaic performance characterization

Solar cell performance was measured using a solar simulator (Oriel, 450 W Xenon, AAA class). The light intensity was calibrated with a Si reference cell (KG3, Newport). Current-voltage (*I-V*) characteristics were measured by applying external voltage bias and detecting the current response with Keithley 2400 digital source meter at room temperature in air under AM1.5 G 1 sun illumination. The voltage scan rate, scan steps, and delay time were 100 mV s$^{-1}$, 10 mV, and 100 ms, respectively. The active area of the devices was defined by using a black metal mask with an aperture area of 0.0804 cm$^2$. No light soaking or voltage bias was applied before the measurement. The modules were measured with a scanning step of 50 mV and a delay time of 200 ms with a metal mask. The low-temperature processed modules were characterized by a solar simulator (Oriel, Xenon, AAA class, 94123 A) and digital source meter (Keithley 2420). The scan step and the delay time were 80 mV and 10 ms, respectively. External quantum efficiency was characterized by using an IQE200B (Oriel) without bias light.

### Stability tests

The stability of the devices was determined at maximum power point tracking in a sealed container filled with nitrogen flow. The light intensity from the LED lamp used in the stability measurement system was 100 mW cm$^{-2}$, which was checked with a reference Si photodiode placed in the container. *J-V* curves were recorded on an electronic system using 22-bit delta-sigma analogic to digital converter with 25 mV s$^{-1}$ of scan rate and 5 mV of step size.

### Film characterization

The absorbance of the films was measured using a UV-vis spectrophotometer (PerkinElmer spectrophotometer). Photoluminescence measurements were performed using a Fluorolog-3 (Horiba Scientific) with an excitation source of 450 nm. Transient PL decay was measured using Fluorolog TCSPC with an excitation wavelength of 635 nm. Temperature-dependent PL measurements were conducted by loading the perovskite films into a cryostat chamber in the temperature range of 80 K to 300 K. X-ray diffraction (XRD) measurement was done in a range of $2\theta = 2°$ to 60° using a Bruker D8 Advance diffractometer. The morphology of films was imaged by using a high-resolution scanning electron microscopy (SEM) (ZEISS Merlin). Two-dimensional grazing- incident wide-angle X-ray scattering (GIWAXS) patterns represented in reciprocal lattice space were performed at SPring-8 on beamline BL19B2. The samples were irradiated with X-ray energy of 12.39 keV ($\lambda$ = 1 Å) at incident angles on the order of 0.08°, 0.12°, 0.16°, 0.20°, and 0.40° through a Huber diffractometer. X-ray photoelectron spectroscopy (XPS) measurements were carried out using a photoelectron spectrometer (Kratos Inc., SUPRA). A monochromatic Al-Kα source (1486.6 eV) was used for XPS. PL imaging of large area films was conducted using a commercially available Solar Cell Imaging Test System (K3300, McScience). A 630 nm red LED was used as an excitation source. The PL maps were obtained through a 5-mega-pixel silicon complementary metal oxide semiconductor (CMOS) camera, which detects the luminescence signal.

### Single crystal XRD studies

Bragg-intensities of 3 C, 2H and 6H were collected at room temperature using MoKα radiation on a Rigaku SuperNova dual system diffractometer equipped with an Atlas S2 CCD detector. The datasets were reduced and corrected for absorption, with the help of a set of faces enclosing the crystals as snugly as possible, with the latest available version of *CrysAlis*$^{Pro}$ [44]. The solutions and refinements of the

structures were performed by the latest available version of *ShelXT*[45] and *ShelXL*[46] using *Olex2*[47] as the graphical interface. All non-hydrogen atoms were refined anisotropically using full-matrix least-squares based on $|F|^2$. The hydrogen atoms were placed at calculated positions by means of the "riding" model in which each H-atom was assigned a fixed isotropic displacement parameter with a value equal to 1.2 $U_{eq}$ of its parent C-atom (1.5 $U_{eq}$ for the methyl groups). Crystallographic and refinement data are summarized in Supplementary Table 6. Crystallographic data have been deposited with the Cambridge Crystallographic Data Centre and correspond to the following codes: 3 C (2115270), 2H (2115268) and 6H (2115269). These data can be obtained free of charge via www.ccdc.cam.ac.uk/data_request/cif, or by emailing data_request@ccdc.cam.ac.uk, or by contacting The Cambridge Crystallographic Data Centre, 12 Union Road, Cambridge CB2 1EZ, UK; fax: +44 1223 336033.

### Reporting summary
Further information on research design is available in the Nature Portfolio Reporting Summary linked to this article.

## Data availability
The data supporting the findings of this study are provided in the Supplementary Information/ Source Data file. Additional data are available from the corresponding author on request. Source data are provided with this paper.

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

## Acknowledgements

We thank the Swiss National Science Foundation (SNSF) for financial support.(M.K.N. and P.D.) This project has received funding from the European Union's Horizon 2020 research and innovation programme under Grant Agreement No. 763989 APOLO.(M.K.N.) The GIWAXS measurements were performed at SPring-8 with the approval of the JASRI (proposal no. 2021B1870) and NS thanks the Foundation for Promotion of Material Science and Technology of Japan (MST) for its support.(N.S.) The research is carried out using the equipment of the shared research facilities of HPC computing resources at Lomonosov Moscow State University.(O.A.S.) This Research has been performed as a project NO. KS2422-10 (A Study on the Convergence Materials for Off-Grid Energy Conversion/Storage Integrated Devices) and supported by the Korea Research Institute of Chemical Technology (KRICT). (N.J.J.) This research was supported by the Digital Research Innovation Institution Program through the National Research Foundation of Korea (NRF) funded by Ministry of Science and ICT (RS-2023-00283597).(H.K.) This research was supported by the National Research Council of Science & Technology (NST) grant by the Korea government (MSIT) (No. CCL23281-100).(H.K.)

## Author contributions

H.K. conceived the idea. H.K., S.-M.Y., and B.D. designed the experiments and performed the fabrication and characterization of perovskite films and devices. S.-M.Y. synthesized perovskite single crystals and F.F.T. performed the SCXRD measurement and data processing. N.S. and H.K. performed GIWAXS measurement and data processing. H.K., S.-M.Y., F.F.T., P.S., and J.P. discussed the result of the crystallography. H.K. performed the absorbance, PL decay, temperature-dependent PL measurement and analysis. H.J.Y. and B.S. contributed to the XPS measurement. M.A.S. and O.A.S performed the theoretical modelling and calculation. S.-M.Y., B.-S.K., Y.Y.K., Y.D. performed the fabrication and characterization of perovskite solar modules. H.K., P.J.D. and M.K.N. supervised the project. H.K. and S.-M.Y. wrote the manuscript and N.J.J., P.J.D. and M.K.N. edited it. All authors reviewed the manuscript.

## Competing interests

The authors declare no competing interests.
