## [Peer Review File · Nature Communications]

Shallow-level Defect Passivation by 6H Perovskite Polytype for Highly Efficient and Stable Perovskite Solar CellsEditorial Note: This manuscript has been previously reviewed at another journal that is not operating a transparent peer review scheme. This document only contains reviewer comments and rebuttal letters for versions considered at *Nature Communications*.

REVIEWER COMMENTS

Reviewer #1 (Remarks to the Author):

In this manuscript by Kim et al., the authors reported that a corner-sharing hexagonal (6H) polytypic perovskite can be constructed at the grain boundary of cubic polytype FAPbI₃ by adding an excess PbI₂. The corner-sharing 6H polytype perovskite engineered defects at the interface with cubic polytype FAPbI₃, which facilitated radiative recombination and improved the stability of the perovskite film, confirmed by X-ray diffraction, 2D grazing-incidence wide-angle X-ray scattering, PL and density functional theory calculation. Hence, the power conversion efficiency of 24.13% and 21.92% has been achieved based on perovskite solar cell and module, respectively. Moreover, the PSCs show a long-term operational stability of 1000 hours with a PCE decay less than 10%.

The authors did hard work to revise the manuscript with reflection of reviewers' comments. All of my comments have been fully addressed. Therefore, I recommend the publication of this manuscript in Nature Communications.

Reviewer #3 (Remarks to the Author):

This reviewer acknowledges and appreciates the authors' diligent efforts in revising the manuscript in response to the reviewer's feedback. This reviewer finds no significant issues with the resubmitted manuscript to Nature Communications. However, considering the materials and compositions employed by the authors, I kindly request confirmation that similar results are achieved when utilizing the specified starting materials (FAI + PbI₂ + 20 mol% MAPbCl₃ + 35 mol% MAI). Additionally, as a minor observation, it is suggested that the labeling of Cl⁻ and I⁻ ions in Figure R2 (Supplementary Figure 5 of the revised manuscript) may be incorrect.

Response to Reviewers

We thank the reviewers for their valuable and insightful comments and suggestions. We have carefully addressed each of the reviewers' points, and thoroughly revised the manuscript and supplementary information in accordance with their suggestions. We believe that our work has been strengthened and is now more convincing due to these revisions, Point-by-point responses are provided below.

Reviewer #1 (Remarks to the Author):

In this manuscript by Kim et al., the authors reported that a corner-sharing hexagonal (6H) polytypic perovskite can be constructed at the grain boundary of cubic polytype FAPbI₃ by adding an excess PbI₂. The corner-sharing 6H polytype perovskite engineered defects at the interface with cubic polytype FAPbI₃, which facilitated radiative recombination and improved the stability of the perovskite film, confirmed by X-ray diffraction, 2D grazing-incidence wide-angle X-ray scattering, PL and density functional theory calculation. Hence, the power conversion efficiency of 24.13% and 21.92% has been achieved based on perovskite solar cell and module, respectively. Moreover, the PSCs show a long-term operational stability of 1000 hours with a PCE decay less than 10%.

The authors did hard work to revise the manuscript with reflection of reviewers' comments. All of my comments have been fully addressed. Therefore, I recommend the publication of this manuscript in *Nature Communications*.

Response)

We appreciate your time and effort in reviewing our work and your acknowledgement of our effort is greatly valued.

Reviewer #3 (Remarks to the Author):

This reviewer acknowledges and appreciates the authors' diligent efforts in revising the manuscript in response to the reviewer's feedback. This reviewer finds no significant issues with the resubmitted manuscript to *Nature Communications*. However, considering the

materials and compositions employed by the authors, I kindly request confirmation that similar results are achieved when utilizing the specified starting materials (FAI + PbI₂ + 20 mol% MAPbCl₃ + 35 mol% MACl). Additionally, as a minor observation, it is suggested that the labeling of Cl⁻ and I⁻ ions in Figure R2 (Supplementary Figure 5 of the revised manuscript) may be incorrect.

Response)

We thank the reviewer for acknowledging and appreciating our work and for the valuable comments. In response to the reviewer's request, we fabricated films using the suggested precursor combination to confirm if similar results are achieved compared to those presented in the manuscript. The target solution suggested by the reviewer (FAI + PbI₂ + 20 mol% MAPbCl₃ + 35 mol% MACl) was prepared by dissolving FAI (1.2 M), PbI₂ (1.2 M), PbCl₂ (0.24 M), and MACl (0.66 M) in a mixed solvent comprising DMF:DMSO (8:1, v/v), whereas the control precursor solution with excess PbI₂ was prepared as described in the manuscript: FAI (1.2 M), PbI₂ (1.44 M), and MACl (0.66 M) in DMF:DMSO (8:1, v/v). **The following text and figure has been added to the revised MS and SI:**

In the MS,

We also investigated whether PbCl₂ could replace excess PbI₂ in forming the 6H phase, resulting in an extremely weak diffraction signal of 3C (Supplementary Fig. 7).

In the SI,

The target solution with PbCl₂ was prepared by dissolving FAI (1.2 M), PbI₂ (1.2 M), PbCl₂ (0.24 M), and MACl (0.66 M) in a mixed solvent comprising DMF:DMSO (8:1, v/v), whereas the control precursor solution with excess PbI₂ was prepared as described in the Materials and Methods: FAI (1.2 M), PbI₂ (1.44 M), and MACl (0.66 M) in DMF:DMSO (8:1, v/v). We confirmed that both solutions were well dissolved (Supplementary Fig. 7a).

After spin-coating and thermal annealing, the target film is orange in color, whereas the control film is black (Supplementary Fig. 7b). XRD analysis revealed that the target films have very low diffraction intensity of the 3C phase (Supplementary Fig. 7c), implying significant thermodynamic changes due to the altered composition. This variation may also lead to low reproducibility, with

diffraction patterns showing inconsistent peaks for 6H and PbI_2 (Supplementary Fig. 7d). In contrast, the control films are highly reproducible, showing consistent diffraction patterns for 6H and PbI_2 . The presence of the PbI_2 peak in the target films, despite not using excess PbI_2 , suggests that PbI_2 does not fully react to form the 3C phase and remains as a residue.

Supplementary Fig. 7.

a, Picture of the prepared solutions with excess PbI_2 , control (left) and with PbCl_2 instead of excess PbI_2 , target (right). **b**, The fabricated films using each precursor solution. **c**, The overall XRD patterns of the control and target perovskite films. **d**, XRD patterns of the perovskite films in the low 2θ angle region.

We corrected the labeling in the Supplementary Figure 5 as pointed out by the reviewer, see revised figure below:

Supplementary Fig. 5.

Schematic of proposed mechanism for the formation of 6H polytype by incorporation of MA⁺ and Cl⁻. Doping MA⁺Cl⁻ into FAPbI₃ results in lattice contraction due to the strong dipole moment of MA⁺ and the smaller size of Cl⁻. This presumably promotes folding of octahedra, leading to the formation of face-sharing component in the 6H polytype.

REVIEWERS' COMMENTS

Reviewer #3 (Remarks to the Author):

This reviewer thanks the authors for addressing the comments raised and agrees to publish the manuscript.